# Plasma Membrane H^+^-ATPase *SmPHA4* Negatively Regulates the Biosynthesis of Tanshinones in *Salvia miltiorrhiza*

**DOI:** 10.3390/ijms22073353

**Published:** 2021-03-25

**Authors:** Xiuhong Li, Bin Zhang, Pengda Ma, Ruizhi Cao, Xiaobing Yang, Juane Dong

**Affiliations:** 1College of Forestry, Northwest A&F University, Yangling 712100, China; lixiuhong@nwsuaf.edu.cn; 2College of Life Sciences, Northwest A&F University, Yangling 712100, China; zhangbin351@nwsuaf.edu.cn (B.Z.); pengdama@nwafu.edu.cn (P.M.); caoruizhi1956@sina.com (R.C.); 3College of Applied Engineering, Henan University of Science and Technology, Sanmenxia 472000, China; bing3698@163.com

**Keywords:** *Salvia miltiorrhiza*, plasma membrane H^+^-ATPase, *SmPHA4*, negative regulation, tanshinone

## Abstract

*Salvia miltiorrhiza* Bunge has been widely used in the treatment of cardiovascular and cerebrovascular diseases, due to the pharmacological action of its active components such as the tanshinones. Plasma membrane (PM) H^+^-ATPase plays key roles in numerous physiological processes in plants. However, little is known about the PM H^+^-ATPase gene family in *S. miltiorrhiza* (*Sm*). Here, nine PM H^+^-ATPase isoforms were identified and named *SmPHA1–SmPHA9*. Phylogenetic tree analysis showed that the genetic distance of *SmPHAs* was relatively far in the *S. miltiorrhiza* PM H^+^-ATPase family. Moreover, the transmembrane structures were rich in *SmPHA* protein. In addition, *SmPHA4* was found to be highly expressed in roots and flowers. HPLC revealed that accumulation of dihydrotanshinone (DT), cryptotanshinone (CT), and tanshinone I (TI) was significantly reduced in the *SmPHA4-OE* lines but was increased in the *SmPHA4-RNAi* lines, ranging from 2.54 to 3.52, 3.77 to 6.33, and 0.35 to 0.74 mg/g, respectively, suggesting that *SmPHA4* is a candidate regulator of tanshinone metabolites. Moreover, qRT-PCR confirmed that the expression of tanshinone biosynthetic-related key enzymes was also upregulated in the *SmPHA4-RNAi* lines. In summary, this study highlighted PM H^+^-ATPase function and provided new insights into regulatory candidate genes for modulating secondary metabolism biosynthesis in *S. miltiorrhiza*.

## 1. Introduction

*Salvia miltiorrhiza* Bunge, one of the best-known Chinese traditional herbs, has been widely used to treat various maladies for more than 2000 years [1]. Specifically, *S. miltiorrhiza* has been shown to have good curative effects on cardiovascular and cerebrovascular diseases [2,3]. Therefore, the contents of bioactive compounds are the main quality control indicators for the evaluation of curative effects of *S. miltiorrhiza*. Unfortunately, the huge market demand and low yield of bioactive compounds in cultivated *S. miltiorrhiza* are setbacks for the industrialised production and commercialisation of *S. miltiorrhiza*. Hence, increasing the contents of bioactive compounds in *S. miltiorrhiza* has become increasingly important [4]. Liposoluble tanshinones, such as dihydrotanshinone (DT), cryptotanshinone (CT), tanshinone I (TI), and tanshinone IIA (TIIA), are the main bioactive compounds in *S. miltiorrhiza*, which are regulated by a series of signals [5,6,7,8], key enzymes [9,10,11], regulatory genes [12,13,14,15,16], and transcription factors [17,18,19,20,21,22]. Currently, optimisation of the biosynthesis of specific metabolites using metabolic engineering is one of the most important techniques to regulate plant secondary metabolism [22,23,24], which is also propitious to satisfy the increasing demand for *S. miltiorrhiza* bioactive compounds. Consequently, various biological methods could be employed to improve the content of tanshinones.

H^+^-ATPase is a functional protein widely present in the plasma membrane (PM) and various intimal systems [25,26]. It has been reported that P-type ATPases, such as PM H^+^-ATPase, are defined by an obligatory phosphorylated reaction cycle intermediate [27]. As one of the primary active transporters, PM H^+^-ATPase mediates ATP hydrolysis and then pumps protons out of a cell [28]. The generated electrochemical potential gradient acts as a driving force for a series of secondary transporter and channel proteins to transport various nutrients and ions across the plasma membrane [29]. Therefore, PM H^+^-ATPase plays a key role in numerous physiological processes, such as nutrient uptake, stomata opening [28,29], cytoplasmic pH regulation [30], cell elongation [31], and cell metabolism [25]. According to recent studies, PM H^+^-ATPase-encoding genes from diverse plant species such as *Arabidopsis* [32,33], *Oryza sativa* (rice) [28], *Nicotiana tabacum* [34,35], *Zea mays* (maize) [36], *Sesuvium portulacastrum* [37], *Cucumis sativus* [38], *Lycopersicon esculentum* (tomato) [39,40,41], and *Gossypium hirsutum* (cotton) [42] have been identified and characterised. However, to date, limited information on the PM H^+^-ATPase gene family from *S. miltiorrhiza* is available.

Our recent investigation found that salicylic acid-induced cytosolic acidification promoted the biosynthesis of phenolic acid compounds in *S. miltiorrhiza* cells, by inhibiting the activity of plasma membrane H^+^-ATPase [43], indicating that the PM H^+^-ATPase-encoding genes might be closely related to the synthesis of secondary metabolites. However, these genes in *S. miltiorrhiza* have not been reported, and their functions in the modulation of biosynthesis of secondary metabolites are not clear. In this study, the PM H^+^-ATPase-encoding genes were screened according to the transcriptome and genome databases of *S. miltiorrhiza*. A total of nine PM H^+^-ATPase isoforms in the *S. miltiorrhiza* genome were identified and named *SmPHA1–SmPHA9*. Tissue expression pattern of the *SmPHAs* showed that *SmPHA4* was highly expressed in roots and flowers. We used reverse genetics to evaluate the role of *SmPHA4,* and transgenic *S. miltiorrhiza* hairy roots were successfully obtained. It was found that *SmPHA4* was involved in modulating the biosynthesis of tanshinones in *S. miltiorrhiza*. These results are not only significant in further understanding the function of PM H^+^-ATPase, but also provide a useful target gene for metabolic engineering to regulate secondary metabolism in *S. miltiorrhiza*.

## 2. Results

### 2.1. Bioinformatics Characterization of PM H^+^-ATPase-Encoding Genes in S. miltiorrhiza

Based on our local transcriptome and *S. miltiorrhiza* genome databases, PM H^+^-ATPase-encoding gene families with nine different mRNA sequences were identified through local comparison and screening and designated as *SmPHA1*–*SmPHA9* (Table 1). The coding region lengths of the *SmPHAs* ranged from 1764 to 3783 bp. To further elucidate the biological characteristics, a series of bioinformatics analyses were performed. As shown in the phylogenetic tree (Figure 1A), genetic distance of the *SmPHAs* was relatively far in *S. miltiorrhiza* PM H^+^-ATPase family; however, a crossover was observed with the lower plant *Marchantia polymorpha* and single-celled species *Saccharomyces cerevisiae*, suggesting that *SmPHAs* are indispensable and multifarious during the evolutionary process. Furthermore, multiple sequence alignment revealed that the functional region of the SmPHAs protein sequences exhibited a high similarity with each other (Figure 1B), suggesting that the sequence of the proteins was highly conserved. Using the online transmembrane prediction analysis system, we found that the SmPHAs were rich in transmembrane structures (Figure 1C). When analysed in detail, up to 11 transmembrane structures were found in SmPHA8 and at least two transmembrane structures in SmPHA9.

### 2.2. Tissue Expression Patterns of SmPHAs

To determine the tissue expression patterns of the nine *SmPHAs*, we measured their expression in various *S. miltiorrhiza* tissues, including the roots, stems, young leaves, mature leaves, and flowers. qRT-PCR analysis (Figure 2) showed that the *SmPHAs* were expressed in all tissues; however, their expression levels were the highest in the roots and successively decreased in stems, mature leaves, and young leaves. One interesting finding was that the expression levels of *SmPHA3*, *4*, *5*, and *7* were remarkably high in roots (more than 1000-fold) compared to that of the reference gene (*SmActin*). In addition, the highest expression levels of *SmPHAs* in stems, flowers, and leaves were of *SmPHA5*, *4*, and *1*, indicating that *SmPHAs* have tissue expression diversity in *S. miltiorrhiza*. Moreover, *SmPHA4* was found to be highly expressed in roots and flowers, indicating that it might be involved in the regulation of secondary metabolites.

### 2.3. Identification of SmPHA4 Transgenic S. miltiorrhiza Hairy Roots

To evaluate the regulatory role of *SmPHA4* in *S. miltiorrhiza*, the overexpression vector pK7GW2R-*SmPHA4* and interference vector pK7GWIWG_Ⅱ-*SmPHA4* were constructed and transformed into *S. miltiorrhiza* hairy roots. The transformants identified by red fluorescent protein (RFP) (Figure 3A) and gene-specific primers (Appendix A) were obtained. qRT-PCR analysis was performed to study the expression level of *SmPHA4* in the transformants. Compared with the control (pK7GW2R-EV), the transcriptional expression of *SmPHA4* in *SmPHA4-OE* lines was upregulated, but the magnitude of the increase was not significant (Figure 3B). As the expression of *SmPHA4* was more than 1000-fold higher in roots compared to that of *SmActin* (Figure 2), we speculated that the effect of *SmPHA4* overexpression was not obvious, mainly because of the excessive basic expression. As expected, the expression of *SmPHA4* in *SmPHA4-RNAi* lines was reduced to 7.8% compared with that in the control (pK7GWIWG2_Ⅱ-EV) (Figure 3B), indicating that RNA interference successfully inhibited the transcription level of *SmPHA4*.

### 2.4. SmPHA4 Is Involved in Modulating the Biosynthesis of Tanshinone

After culturing in 6,7-V liquid medium for three weeks, *SmPHA4-OE*, *SmPHA4-RNAi,* and the respective control hairy root lines were harvested to observe the growth phenotypes. It was observed that the hairy roots, cultured-liquid medium and 70% methanol extraction of *SmPHA4-OE* lines were light yellow, indicating that there was no obvious colour change compared to that in the control lines (pK7GW2R-EV) (Figure 4A). However, the hairy roots of *SmPHA4-RNAi* lines were red. Intriguingly, the cultured-liquid medium and extraction both showed a corresponding red colour (Figure 4C). HPLC analysis revealed that the accumulation of dihydrotanshinone (DT), cryptotanshinone (CT) and tanshinone I (TI) was significantly decreased in the *SmPHA4-OE* lines (Figure 4B) but was prominently increased in *SmPHA4-RNAi* lines (Figure 4D) as compared to the respective controls. The results showed that the contents of DT, CT, and TI were in the range of 2.54–3.52, 3.77–6.33, and 0.35–0.74 mg/g in the *SmPHA4-RNAi* lines, respectively, which represent 2.37–3.29, 2.27–3.81, and 1.28–2.67-fold increases, compared to the control. Therefore, our results demonstrated that *SmPHA4* may be involved in the biosynthesis of tanshinone and negatively regulates tanshinone biosynthesis in *S. miltiorrhiza* hairy roots.

### 2.5. SmPHA4 Affected the Expression of Key Enzymes in the Tanshinone Biosynthetic Pathway

To further explore the regulation mechanism of *SmPHA4* in tanshinone biosynthesis, qRT-PCR was used to detect the expression of key enzymes related to the tanshinone biosynthetic pathway. Compared to that in the control, the expression levels of *SmDXS2* and *SmCYP76AH1* were increased, while those of *SmAACT1* and *SmKSL1* were decreased in *SmPHA4-OE* lines (Figure 5A). However, expression levels of all the four were increased in the *SmPHA4-RNAi* lines (Figure 5B). These results further suggested that *SmPHA4* may be involved in modulating tanshinone biosynthesis in *S. miltiorrhiza* by affecting the expression of *SmDXS2*, *SmAACT1*, *SmKSL1,* and *SmCYP76AH1*.

## 3. Discussion

### 3.1. Nine PM H^+^-ATPase Isoforms Are Present in S. miltiorrhiza

It has been found that PM H^+^-ATPase in plants is encoded by a multigene family [28,32]. For example, bioinformatics analysis shows 11 PM H^+^-ATPase isoforms in *Arabidopsis* (*AHA1–AHA11*) [32,33], ten in *Oryza sativa* (rice) (*OsA1*–*OsA10*) [28], and nine in *Nicotiana tabacum* (*PMA1–PMA9*) [34,35]. In this study, nine PM H^+^-ATPase–encoding genes (*SmPHA1–SmPHA9*) were identified (Table 1) from *S. miltiorrhiza* transcriptome and genome databases, providing evidence that PM H^+^-ATPase is encoded by a multigene family. Given the high similarity of the protein sequences (Figure 1B) and abundant transmembrane structures of the *SmPHAs* (Figure 1C), we speculated that PM H^+^-ATPase genes from *S. miltiorrhiza* could have a potential function for transmembrane transport. Based on the high expression of *SmPHA4* in roots (Figure 2), we deduced that *SmPHA4* might be involved in the process of *S. miltiorrhiza* secondary metabolism, as the main bioactive compounds were mainly accumulated in the roots. To explore its function, *SmPHA4* was successfully isolated and genetically transformed into *S. miltiorrhiza* hairy roots using genetic engineering techniques.

### 3.2. SmPHA4 Negatively Regulated the Biosynthesis of Tanshinone and May Be a Candidate Regulator of Tanshinone Metabolites in S. miltiorrhiza

Tanshinones, widely accumulated in the roots of *S. miltiorrhiza* [2,44], are diterpenoid compounds that have significant anti-aging, anti-inflammatory, and antioxidant activities [45,46]. Most of the tanshinone biosynthesis key enzyme genes, such as *SmDXS2*, *SmAACT1*, *SmKSL1*, and *SmCYP76AH1*, are involved in the tanshinone metabolism pathway [11,13]. *AACT*, the first enzyme in the terpene synthesis pathway, catalyses acetyl-CoA to acetoacetyl-CoA [11]. It has been reported that the expression level of *AACT* is significantly correlated with tanshinone production [47]. *SmDXS2* has been identified as a potential key enzyme in the pathway involved in targeted metabolic engineering to increase the accumulation of tanshinone in *S. miltiorrhiza* hairy roots [48,49]. *CYP76AH1* has been demonstrated to catalyse the turnover of miltiradiene in tanshinone biosynthesis [15]. Moreover, overexpression of *SmKSL* increased the yield of total tanshinone by a maximum of 2.7-fold in the transgenic *S. miltiorrhiza* hairy roots than in wild-type control lines [12].

In this study, we discovered that the phenotypes of *SmPHA4-OE* hairy root lines showed no obvious colour change compared to those of the control (Figure 4A). However, *SmPHA4-RNAi* hairy root lines were dark red, and the liquid medium and 70% methanol extraction both exhibited the same colour (Figure 4C), indicating that *SmPHA4* had a significant effect on the phenotype of *S. miltiorrhiza* hairy roots. It has been reported that gradation of the colour of radix surface is positively correlated with tanshinone compounds in *S. miltiorrhiza*. The higher the tanshinone content, the darker the root colour [50]. Thus, the *SmPHA4-RNAi* hairy root lines should contain a higher yield of tanshinone. As expected, HPLC analysis revealed that the accumulation of DT, CT, and TI significantly decreased in the *SmPHA4-OE* lines but prominently increased in the *SmPHA4-RNAi* lines ranging from 2.54 to 3.52, 3.77 to 6.33, and 0.35 to 0.74 mg/g, respectively (Figure 4). These results further supported the correlation between phenotype and tanshinone content. Additionally, expression levels of *SmDXS2*, *SmAACT1*, *SmKSL1*, and *SmCYP76AH1* were increased in the *SmPHA4-RNAi* lines (Figure 5B). Therefore, *SmPHA4* was speculated to be a negative regulator of the tanshinone biosynthetic pathway, which could negatively regulate the biosynthesis of tanshinone by modulating the expression of tanshinone synthesis-related enzyme genes, including *SmDXS2*, *SmAACT1*, *SmKSL1*, and *SmCYP76AH1*.

PM H^+^-ATPase is encoded by a family of genes, and different sub-isoforms have various characteristics on the physiological function and regulation [51]. Overexpression of *SpAHA1* confers salt tolerance on transgenic *Arabidopsis* [37]. Three members of *Arabidopsis* H^+^-ATPases, such as *AHA6*, *AHA8*, and *AHA9*, are redundantly involved in the generation of the electrochemical potential gradient and are essential for regulating pollen tube growth [52]. Silencing *OsA2* has been shown to not only affect grain yield and shoot growth, but also decrease nitrogen concentration in *Oryza sativa* [53]. Moreover, *OsA7* is involved in blue light-induced stomatal opening of dumbbell-shaped guard cells in monocotyledon species [54]. PM H^+^-ATPase, a primary transporter, plays a central role in transport across the plasma membrane [37]. In this study, we identified a new function of the PM H^+^-ATPase, as SmPHA4 was found to be a negative regulator of the tanshinone biosynthetic pathway and played a role in the regulation of tanshinone biosynthesis.

Dry roots of *S. miltiorrhiza* are widely used to produce a variety of traditional Chinese medicines [1] to treat various maladies, such as cardiovascular disease, low blood circulation, inflammation and angina pectoris [3,33]. Lipid-soluble tanshinones, such as CT, DT, and TI, are important pharmacologically active compounds in *S. miltiorrhiza*. With increasing market demand, higher yield and quality of *S. miltiorrhiza* have become a necessity [55]. Therefore, the yield of tanshinones from *S. miltiorrhiza* must be increased for clinical value. Metabolic engineering can regulate secondary metabolism of *S. miltiorrhiza* through genetic modification of biosynthetic pathways [56]. Currently, most reports have focused on targeted metabolic engineering by key enzymes [8,49,57] or transcription factors [22,58]. Table 2 shows several genes (including *SmGRAS3*, *SmJAZ8*, and *SmKSL*) and transcription factors (including *SmWRKY2*, *SmMYB36*, *SmERF1L1*, *SmERF115*, *SmMYB98*, and *SmWRKY1*) that regulate the tanshinones biosynthesis. The yield of total tanshinones in *SmPHA4* transgenic *S. miltiorrhiza* hairy roots (*SmPHA4-RNAi* lines) obtained in our study was slightly lower than that of *SmERF115*, *SmMYB98* and *SmWRKY1* transgenic lines but was significantly higher than that of *SmGRAS3*, *SmJAZ8*, *SmKSL*, *SmWRKY2*, *SmMYB36*, and *SmERF1L1* transgenic lines, indicating that *SmPHA4* exhibited relatively strong ability to regulate tanshinone biosynthesis and could be a candidate regulator of the accumulation of tanshinone metabolites. *SmPHA4*, one of the genes encoding PM H^+^-ATPase, is neither a key gene nor a transcription factor of biosynthetic pathways; however, we found that it could play an important regulatory role in the secondary metabolites of *S. miltiorrhiza* (Figure 4). Therefore, our results were not only important in uncovering the function of PM H^+^-ATPase but also provided new insights into regulatory candidate genes for secondary metabolism in *S. miltiorrhiza*.

## 4. Materials and Methods

### 4.1. Plant Materials

Different tissue samples of *S. miltiorrhiza*, including roots, stems, young leaves, mature leaves, and flower tissue, were collected from two-year-old *S. miltiorrhiza* plants, which were grown in the medicinal botanical garden of Northwest A&F University, for RNA isolation to analyse the tissue expression patterns of PM H^+^-ATPase–encoded genes. *S. miltiorrhiza* sterile plantlets were cultured in solid 1/2 MS medium with a 16 h light/8 h dark cycle at 25 °C. Using previously published methods [62], *S. miltiorrhiza* hairy roots were derived from sterile leaves infected with *Agrobacterium rhizogenes* strain ATCC15834. Fresh hairy roots of transformants and controls were cultured in 6,7-V liquid medium supplemented with 3% sucrose (50 mL in 150 mL Erlenmeyer flasks) and cultivated on a rotary shaker at 125 rpm and 25 °C in the dark. The hairy root lines (0.3 g) were sub-cultured every 3 weeks.

### 4.2. Bioinformatics Analysis

A local transcription database of *S. miltiorrhiza* was built (NCBI SRA database, login number:SRX1423774), as previously reported [63]. To verify all the potential PM H^+^-ATPase–encoded genes in *S. miltiorrhiza*, *Arabidopsis* PM H^+^-ATPase genes were queried in BLAST searches against our local *S. miltiorrhiza* transcriptome databases and *S. miltiorrhiza* genomic database (http://www.ndctcm.org/shujukujieshao/2015-04-23/27.html (accessed on 31 July 2018)). A phylogenetic tree was constructed based on the protein sequences of the PM H^+^-ATPase family of *S. miltiorrhiza* from the *S. miltiorrhiza* genome database and some model species, such as *Arabidopsis*, fungi and bryophytes, from the NCBI database. Systematic evolution analysis was carried out using the neighbour-joining method employing MEGA7.0 software. Multiple sequence alignment was performed using BioEdit V7.0 software for local BLAST and bidirectional BLAST analysis. Transmembrane structure was predicted using the TMHMM Server v.2.0 software (http://www.cbs.dtu.dk/services/TMHMM/ (accessed on 3 March 2020)).

### 4.3. Plant RNA Isolation and qRT-PCR Analysis

Total RNA was extracted from five *S. miltiorrhiza* tissues and transgenic *S. miltiorrhiza* hairy roots and then reverse-transcribed into cDNA using the RNAprep pure plant kit and Reverse Transcriptase Kit (TransGen Biotech Co., Ltd., Beijing, China), following the manufacturer’s instructions. Real-time reverse transcription PCR (qRT-PCR) was performed using 10 μL 2×Transstart^®^ Tip Green qPCR SuperMix, 0.5 μL/0.5 μL (10 μM) Primer-For/Rev, 1 μL cDNA template and 8 μL ddH_2_O. PCR was performed as follows: step 1, 50 °C for 5 min; step 2, 95 °C for 30 s; step 3, 95 °C for 10 s, 58 °C for 15 s, 72 °C for 15 s, 40 cycles. The qRT-PCR cycling program was performed on a real-time PCR system (CFX96, Bio-Rad, Hercules, CA, USA). Relative gene expression was calculated using the 2^-^^△Ct^ method, where △Ct = Ct*_target gene_*-CT*_actin_*. The *S. miltiorrhiza* actin gene [64] was used as the internal reference gene. All the primer sequences used for qRT-PCR analyses are listed in Appendix A.

### 4.4. Plant Expression Vector Construction

The gateway method was used to construct *SmPHA4*-overexpression vectors and RNA interference vectors. To construct *SmPHA4*-overexpression vectors, *SmPHA4* was amplified with primers SmPHA4-attB-for/SmPHA4-attB-Rev (Appendix A) and then successively inserted into the entry vector pDONR207 and the overexpression vector pK7GW2R by using the BP Clonase Enzyme Kit and LR Clonase Enzyme Kit according to the manufacturer’s instructions (Invitrogen, Waltham, MA, USA). An pK7GW2R empty vector without the *SmPHA4* gene was used as a control for the *SmPHA4-OE* lines. Concurrently, using on-line SIRNA design tools such as siDirect version 2.0 (http://sidirect2.rnai.jp/ (accessed on 2 September 2018)) and Designer of Small Interfering RNA (http://biodev.extra.cea.fr/DSIR/DSIR.html (accessed on 2 September 2018)) to search for suitable RNA interference fragments, a piece of RNA interference fragments of 300 bp in length was found in the coding sequence (CDS) region of *SmPHA4*. The reference fragment for RNA interference was a 300 bp gene sequence from an RFP protein-encoding gene and used as a control for the *SmPHA4-RNAi* lines. Similarly, to construct an RNAi vector, the interference fragments were amplified with RNA interference primers (Appendix A) and then inserted into the entry vector pDONR 207 by BP Clonase Enzyme Kit and in the RNAi vector pK7GWIWG2_II by LR Clonase Enzyme Kit. The resulting pDONR207-*SmPHA4*, pK7GW2R-*SmPHA4*, recombinant interference vectors (RNAi-*SmPHA4*) and recombinant interference vector containing unrelated genes (pK7GWIWG2_II-RNAi-CK) were sequenced by TSINGKE Biological Technology Co., Ltd. (Xi’an, China). All the vectors contained red fluorescent protein (RFP) for identification of fluorescence.

### 4.5. Genetic Transformation and Verification

All the recombinants were transformed into *Agrobacterium rhizogenes* strain ATCC15834, and the positive clones were used for infecting *S. miltiorrhiza* sterile leaves to induce *S. miltiorrhiza* hairy roots using a method described previously [62]. The positive transgenic lines were screened for fluorescence identification and for genomic DNA identification using specific primers such as *rolB*, *rolC*, *pK7-NPTII*, and *p35S+*RNAi*SmPHA*4-attB-Rev/*RNAi-dsRed*-Rev. RFP expression was observed under a fluorescence microscope (Leica DM5000 B, Wetzlar, Germany). Genomic DNA was isolated from fresh hairy roots using the TIANamp Genomic DNA Kit (TIANGEN Biotech Co., Ltd., Beijing, China), following the manufacturer’s instructions. Positive transgenic and control hairy root lines, harvested after 3 weeks, were used for qRT-PCR and high-performance liquid chromatography (HPLC) analyses.

### 4.6. Determination of Tanshinone Contents with HPLC

The contents of secondary metabolites in transgenic hairy roots were analysed by HPLC. The harvested *S. miltiorrhiza* hairy roots were dried in an oven at 42 °C until the weight remained constant and then ground into powder using a mortar. The powder samples (20 mg) were subjected to ultrasonic-assisted extraction with 4 mL of 70% methanol for 45 min in 10 mL centrifuge tubes and then kept overnight. The extracts were centrifuged at 12,000× *g* for 10 min and the supernatants were subsequently filtered through 0.22 μm Millipore filters. The filtrates were analysed with HPLC (Waters 1525, Milford, MA, USA), using a UV dual-absorbance detector (Waters 2487) and a reversed-phase C18 column (250 mm × 4.6 mm, 5 μm, Shimadzu, Kyoto, Japan). The contents of DT, CT, and TI were determined under the following chromatographic conditions: mobile phase, acetonitrile (A) and 0.02% phosphoric acid (B); flow rate, 1 mL/min; sample injection volume, 10 µL; detection wavelength, 270 nm; and column temperature, 30 °C. The gradient elution approach was set as follows: t = 0 min, 5% A; t = 5 min, 20% A; t = 16 min, 35% A; t = 31 min, 60% A; t = 41 min, 70% A; t = 45 min, 100% A; t = 55 min, 5% A; t = 60 min, 5% A. According to the peak area obtained, the relevant product concentration was calculated using the following standard curve Equations (1)–(3). Standards of DT, CT, and TI were purchased from Century Aoko Biotechnology Co. LTD (Bingjing, China):*Y*_1_ = 38,341.1739 *X*_1_ + 1302.107252 R^2^ = 0.999578(1)
*Y*_2_ = 55,880.55075 *X*_2_ +7021.294612 R^2^ = 0.998929(2)
*Y*_3_ = 23,639.17824 *X*_3_ + 26,898.98745 R^2^ = 0.998700(3)
where *Y*_1_, *Y*_2_, and *Y*_3_ are the contents of DT, CT, and TI, respectively; *X*_1_, *X*_2_, and *X*_3_ are the peak areas of DT, CT, and TI, respectively.

### 4.7. Statistical Analyses

All the experiments were performed with at least three biological replicates, and statistical analysis was carried out using GraphPad Prism 5 (GraphPad Software, Inc., San Diego, CA, USA). The results were expressed as mean ± standard deviation and were analysed employing the t test.

## 5. Conclusions

In this study, nine PM H^+^-ATPase isoforms in the *S. miltiorrhiza* genome were identified and named as *SmPHA1–SmPHA9*, respectively. Intriguingly, *SmPHA4* was not only highly expressed in roots and flowers but also negatively regulated the biosynthesis of tanshinones in *S. miltiorrhiza* by affecting the expression of biosynthetic genes, including *SmDXS2*, *SmAACT1*, *SmKSL1*, and *SmCYP76AH1*. *SmPHA4* has a potential role in the development of biotechnological strategies to improve tanshinone production. Accumulation of DT, CT, and TⅠ prominently increased in the *SmPHA4-RNAi* lines ranging from 2.54 to 3.52, 3.77 to 6.33, and 0.35 to 0.74 mg/g, respectively. Therefore, the transgenic *S. miltiorrhiza* hairy roots of *SmPHA4* provided a direct proof that PM H^+^-ATPase is involved in the regulation of secondary metabolic processes and that it plays a physiological role in *S. miltiorrhiza*.

## Figures and Tables

**Figure 1 ijms-22-03353-f001:**
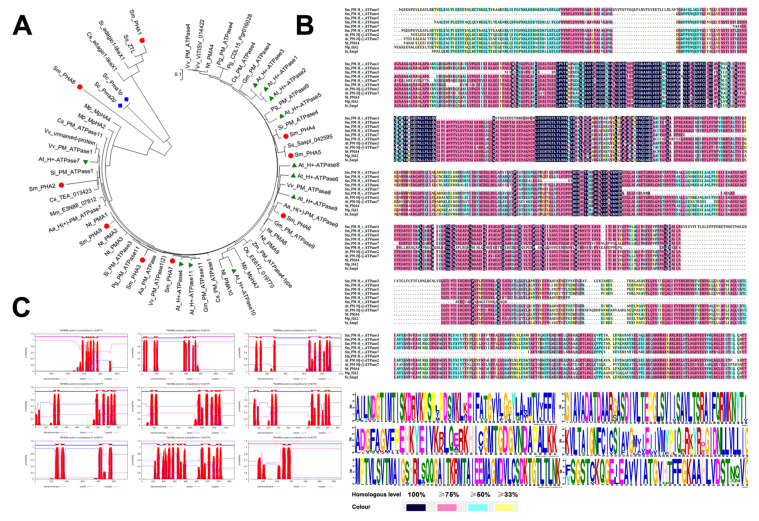
(**A**) Phylogenetic tree showing the relationship between *SmPHAs* and other PM H^+^-ATPase genes. Phylogenetic tree was constructed by the neighbor-joining methods in MEGA7.0 software based on amino acid sequence alignment and demonstrated the relationship of the PM H^+^-ATPase genes from *S. miltiorrhiza* (Sm), *Arabidopsis thaliana* (At), *Nicotiana tabacum* L. (Nt), *Saccharomyces cerevisiae* (Sc), *Vitis vinifera* (Vv), *Oryza sativa* (Os), *Zea mays* (Zm), *Marchantia polymorpha* (Mp), *Salvia splendens* (Ss), *Sesamum indicum* (Si), *Artemisia annua* (Aa), *Punica granatum* (Pg), *Glycine max* (Gm), *Mikania micrantha* (Mm), and *Camellia sinensis* (Cs). The red five-pointed stars, black circles, blue squares, and green triangles indicate *S. miltiorrhiza*, *Arabidopsis thaliana*, *Saccharomyces cerevisiae*, and *Marchantia polymorpha*, respectively. (**B**) Multiple alignment of SmPHAs with related PM H+-ATPase proteins from other plant species. Protein sequence alignment of PM H+-ATPase proteins were from *S. miltiorrhiza*, *Arabidopsis thaliana*, *Nicotiana tabacum* L., *Salvia splendens*, and *Marchantia polymorpha*. Black boxes indicate identical residues; pink boxes, blue boxes, and yellow boxes successively indicate identical residues for at least 75%, 50%, and 33%. (**C**) Transmembrane structure prediction of PM H^+^-ATPase family in *S. miltiorrhiza*. The transmembrane structures were predicted by TMHMM Server v.2.0 software (http://www.cbs.dtu.dk/services/TMHMM/ (accessed on 3 March 2020)).

**Figure 2 ijms-22-03353-f002:**
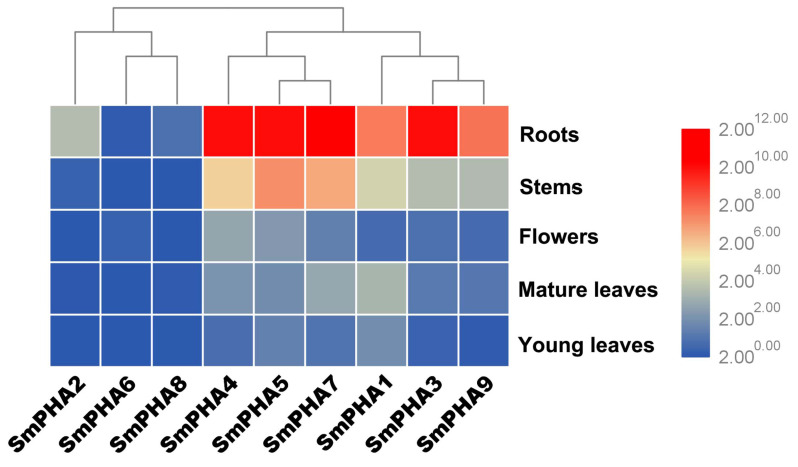
Expression patterns of *SmPHAs* in different tissues. Each tissue was collected from several individual two-year-old *S. miltiorrhiza* plants cultured in nature. *SmActin* was used as an internal control.

**Figure 3 ijms-22-03353-f003:**
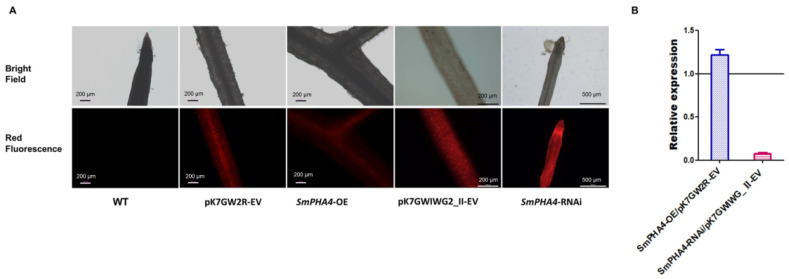
(**A**) Fluorescence observation of transgenic *S. miltiorrhiza* hairy roots. The transgenic lines were obviously showed red fluorescence. (**B**) Transcriptional expression analysis in *S. miltiorrhiza* transgenic hairy root lines. The transcriptional expression levels of *SmPHA4* in *SmPHA4-OE* and *SmPHA4-RNAi* transgenic hairy root lines were detected by qRT-PCR. The average transcriptional expression level of each empty vector was set to 1. *SmActin* was used as the internal reference gene. Error bars represent the SD of three biological replicates.

**Figure 4 ijms-22-03353-f004:**
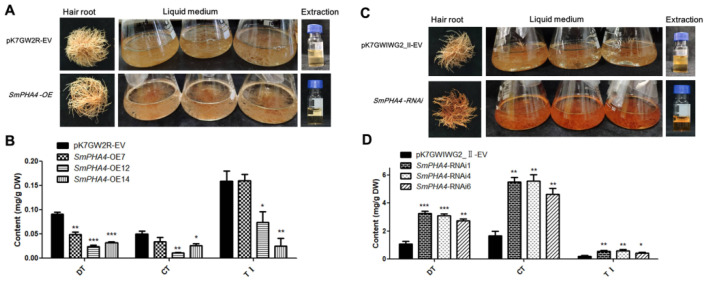
Analysis of the tanshinones in the *SmPHA4-OE* and *SmPHA4-RNAi* transgenic *S. miltiorrhiza* hairy roots. (**A**,**C**) The phenotype, liquid medium, and extraction of the *SmPHA4-OE* and *SmPHA4-RNAi* transgenic *S. miltiorrhiza* hairy roots. (**B**,**D**) The contents of tanshinones in the *SmPHA4-OE* and *SmPHA4-RNAi* transgenic *S. miltiorrhiza* hairy roots were detected by HPLC. The hairy roots were obtained after cultured for 3 weeks. Error bars represent the SD of three biological replicates. *, *p* < 0.05; **, *p* < 0.01; ***, *p* < 0.001. DT, dihydrotanshinone; CT, cryptotanshinone; TI, tanshinone I.

**Figure 5 ijms-22-03353-f005:**
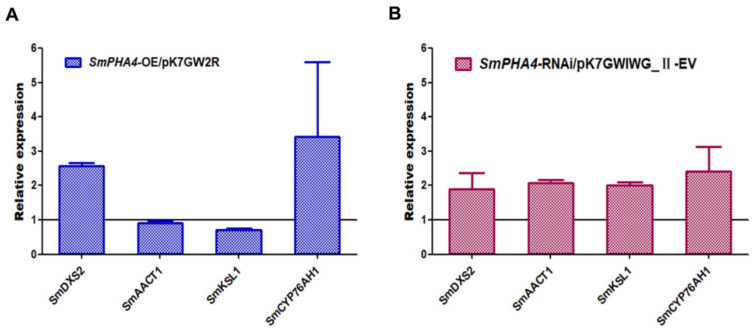
Transcriptional expression analysis in *S. miltiorrhiza* transgenic hairy root lines. The transcriptional expression levels of tanshinone biosynthesis pathway genes were detected by qRT-PCR. The average transcriptional expression level of each gene in the two control hairy root lines was set to 1. *SmActin* was used as the internal reference gene. Error bars represent the SD of three biological replicates. (**A**) The transcriptional expression levels of tanshinone synthesis-related enzyme genes in *SmPHA4-OE* transgenic hairy root lines. (**B**) The transcriptional expression levels of tanshinone synthesis-related enzyme genes in *SmPHA4-RNAi* transgenic hairy root lines.

**Table 1 ijms-22-03353-t001:** Statistics of plasma membrane H^+^-ATPase in *Salvia miltiorrhiza* genome database.

PM H^+^-ATPase	Genome Database Number	Forecast CDS Size	Gene Name
PM H^+^-ATPase 1	C220613.11.2	3408 bp	*SmPHA1*
PM H^+^-ATPase 2	scaffold5589.3	3783 bp	*SmPHA2*
PM H^+^-ATPase 3	C222433.30	3027 bp	*SmPHA3*
PM H^+^-ATPase 4	scaffold1405.24	2523 bp	*SmPHA4*
PM H^+^-ATPase 5	C222429.17	2313 bp	*SmPHA5*
PM H^+^-ATPase 6	scaffold3077.31	2814 bp	*SmPHA6*
PM H^+^-ATPase 7	scaffold376.37	1764 bp	*SmPHA7*
PM H^+^-ATPase 8	scaffold10570.1	2544 bp	*SmPHA8*
PM H^+^-ATPase 9	scaffold2507.1	1962 bp	*SmPHA9*

**Table 2 ijms-22-03353-t002:** Comparisons of tanshinone accumulation of regulatory genes and transcription factors in *S. miltiorrhiza*
^a^.

Genes	Raw Material	Strategy	Tanshinone, mg/g				Reference
			DT	CT	TI	TTAs	
*SmPHA4*	*S. miltiorrhiza* hairy roots	Suppression	2.54–3.52	3.77–6.33	0.35–0.74	6.66–10.59	Present study
*SmGRAS3*	*S. miltiorrhiza* hairy roots	Overexpression	0.1–0.2	0.2–0.4	0.8–0.9	1.1–1.5	[59]
*SmJAZ8*	*S. miltiorrhiza* hairy roots	Suppression	0.4–0.7	0.3–0.4	0.8–1.2	1.5–2.3	[16]
*SmKSL*	*S. miltiorrhiza* hairy roots	Overexpression	1.1–1.2	0.6–1.1	0.5–0.8	2.2–3.1	[12]
*SmWRKY2*	*S. miltiorrhiza* hairy roots	Overexpression	0.4–1.1	1.0–1.3	1.7–2.0	2.1–4.4	[58]
*SmMYB36*	*S. miltiorrhiza* hairy roots	Overexpression	0.2–0.4	0.1–1.1	0.4–0.7	0.7–2.8	[17]
*SmERF1L1*	*S. miltiorrhiza* hairy roots	Overexpression	1.5–2.1	3.5–5.2	1.3–1.9	6.3–9.2	[10]
*SmERF115*	*S. miltiorrhiza* hairy roots	Suppression	2.1–3.8	3.9–6.2	0.8–2.0	6.8–12	[56]
*SmMYB98*	*S. miltiorrhiza* hairy roots	Overexpression	3.4–5.8	1.5–3.4	2.5–6.1	7.4–15.3	[60]
*SmWRKY1*	*S. miltiorrhiza* hairy roots	Overexpression	2.0–3.0	2.4–3.8	4.5–6.4	8.9–13.2	[61]

^a^ Abbreviations: DT, dihydrotanshinone; CT, cryptotanshinone; TI, tanshinone I; TTAs, the summed content of DT, CT and TI.

## Data Availability

Data is contained within the article or Appendix A.

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
