# Peer review of "Plasma Membrane H+-ATPase SmPHA4 Negatively Regulates the Biosynthesis of Tanshinones in Salvia miltiorrhiza"

_ijms, 2021, doi:10.3390/ijms22073353_

Round 1

Reviewer 1 Report

This is a most intriguing paper because it makes you think about the problems of extensive gene redundancy and negative regulation always more difficult to elucidate than positive! And, specifically the roles of multiple PM H ATPases (no less than nine in Salvia!) that involve generation of the electrochemical potential gradient and its many consequences.

Overall, it all hangs together nicely with some well-designed experiments

Generally the argument for negative regulation of tanshinone biosynthesis by the plasma membrane H+ATPase SmPHA4 Is clearly laid out: it was increased in the SmPHA4-RNAi lines. The one exception which needs clarification is the sentence on line 193-4: [repeated on lines 332-333]: SmPHA4 “could negatively regulate the biosynthesis of tanshinone by elevating the expression of tanshinone synthesis-related enzyme genes”  

That seems to convey the opposite sense....perhaps “elevating” is the wrong word? On the other hand, end-product inhibition via allosteric enzyme inhibition is probably involved and might be discussed.

Author Response

Point 1: The “elevating” from the sentence on line 193-4 (repeated on lines 332-333) is the wrong word?

Response 1: Thank you very much for the positive and constructive comments and suggestions on our manuscript, we have made revision which marked in red in the paper.

Point 2: The end-product inhibition via allosteric enzyme inhibition is probably involved and might be discussed.

Response 2: Thank you very much for the positive and constructive comments and suggestions on our manuscript. We have studied the comment carefully.  Unfortunately, we have not found the literature supporting the relationship between allosteric enzymes and the accumulation of tanshinones. It is unclear that how the allosteric effect regulate the biosynthesis of secondary metabolites in Salvia miltiorrhiza. As one of the primary active transporters, plasma membrane H+-ATPase mediates ATP hydrolysis and pumps protons out of a cell, ant then induces a series of physiological and biochemical reactions, including the change of cytoplasmic pH and ions transport. In this paper, we found that SmPHA4 negatively regulated the biosynthesis of tanshinones in S. miltiorrhiza by affecting the expression of biosynthetic genes, but we are still unclear the mechanism of SmPHA4 regulating tanshinone synthesis. We will explore the mechanism of SmPHA4 regulating tanshinone synthesis in the further research.

Reviewer 2 Report

This work identifies nine plasma membrane ATPases of Salvia miltiorrhiza Bunge, and highlights a link between ATPase activity and tanshinone biosynthesis. The manuscript is well written, is well structured, with high quality figures and respect scientific formats. The manuscript is easily read and is not too long. This is a valuable manuscript describing a functional approach using appropriate methodologies for studying the transcription and the role of each gene by overexpression and RNA silencing. Figures, tables and supplementary data are appropriate. The information presented is new. The conclusions are supported by the data. The results are sufficiently new and of definite scientific interest to be published. This manuscript can be published as it stands in International Journal of Molecular Sciences.

Author Response

 Thank you very much for the positive and constructive comments and suggestions on our manuscript. We would like to express our great appreciation to you for comments on our paper.